# Community Health Impacts of the Trident Copper Mine Project in Northwestern Zambia: Results from Repeated Cross-Sectional Surveys

**DOI:** 10.3390/ijerph17103633

**Published:** 2020-05-21

**Authors:** Astrid M. Knoblauch, Andrea Farnham, Hyacinthe R. Zabré, Milka Owuor, Colleen Archer, Kennedy Nduna, Marcus Chisanga, Leonard Zulu, Gertrude Musunka, Jürg Utzinger, Mark J. Divall, Günther Fink, Mirko S. Winkler

**Affiliations:** 1Swiss Tropical and Public Health Institute, P.O. Box, 4002 Basel, Switzerland; andrea.farnham@swisstph.ch (A.F.); raogohyacinthe.zabre@swisstph.ch (H.R.Z.); juerg.utzinger@swisstph.ch (J.U.); guenther.fink@swisstph.ch (G.F.); mirko.winkler@swisstph.ch (M.S.W.); 2University of Basel, P.O. Box, 4001 Basel, Switzerland; 3Shape Consulting, P.O. Box 602, St Peter Port GY1, Guernsey, UK; mowuor@shapeconsulting.org (M.O.); mdivall@shapeconsulting.org (M.J.D.); 4Pollution Research Group, Department of Engineering, University of KwaZulu-Natal, 4041 Durban, South Africa; archerc@ukzn.ac.za; 5Nvumabaranda Public Health Services, Ndola, Zambia; kennedy.nduna@yahoo.com; 6First Quantum Minerals Limited, Lusaka, Zambia; marcus.chisanga@fqml.com (M.C.); gertrude.musunka@fqml.com (G.M.); 7Independent Researcher, Lusaka, Zambia; leonardzulu@gmail.com

**Keywords:** health impact assessment, malaria, mining, monitoring and evaluation, nutrition, schistosomiasis, soil-transmitted helminths, syphilis, Zambia

## Abstract

The application of a health impact assessment (HIA) for a large-scale copper mining project in rural Zambia triggered the long-term monitoring and evaluation of determinants of health and health outcomes in communities living in proximity to the mine. Three consecutive cross-sectional surveys were conducted at intervals of four years; thus, at baseline (2011), four (2015) and eight (2019) years into the project’s development. Using the same field and laboratory procedures, the surveys allowed for determining changes in health indicators at the household level, in young children (<5 years), school attendees (9–14 years) and women (15–49 years). Results were compared between communities considered impacted by the project and communities outside the project area (comparison communities). The prevalence of *Plasmodium falciparum* infection increased in both the impacted and comparison communities between 2011 and 2019 but remained consistently lower in the impacted communities. Stunting in children < 5 years and the prevalence of intestinal parasite infections in children aged 9–14 years mostly decreased. In women of reproductive age, selected health indicators (i.e., anaemia, syphilis, underweight and place of delivery) either remained stable or improved. Impacted communities generally showed better health outcomes than comparison communities, suggesting that the health interventions implemented by the project as a consequence of the HIA have mitigated potential negative effects and enhanced positive effects. Caution is indicated to avoid promotion of health inequalities within and beyond the project area.

## 1. Introduction

The economy of Zambia depends heavily on the extraction of natural resources, particularly copper [1]. Trade-offs between economic development and potential positive or negative consequences on health and wellbeing resulting from these mining activities must be considered [2]. While mining of copper and other metals can contribute substantially to government revenue and create employment opportunities, the negative effects on the environment and human health, both equally important resources of the country, can be detrimental and warrant protection [3,4].

The setting of the current study exemplifies the complex nexus between natural resource extraction, the environment and human health. Copper deposits in the Kalumbila district of the North-Western province of Zambia have been exploited since 2011. The Trident Copper Mine project (hereafter, “project”), operated by First Quantum Minerals Limited (FQML), is a greenfield copper mine development located in a previously underdeveloped, rural area [5]. Rapid transformation to the local environment, economy, employment opportunities, infrastructure and social fabric were expected to occur from the mine development. FQML commissioned a health impact assessment (HIA) in 2010, prior to the development of the mine. The goal was to anticipate and manage potential health impacts on communities in proximity to the project [6,7,8]. The HIA applied the determinants of a health model—comprising of the social, economic and physical environment, as well as individual characteristics and behaviours—to cover the full range of potential direct and indirect health effects of the project on the population [9,10]. The specific objectives of the HIA were to (i) prevent negative health impacts; (ii) promote health opportunities; (iii) develop a community health management and monitoring plan; and (iv) act on monitoring data to adjust health interventions over time [10]. The monitoring plan included both collection of routine health information system data as well as periodic, cross-sectional survey data for indicators that were considered potential determinants for human health within the HIA.

Here, we present the changes in selected community health indicators that are associated with the project using monitoring data from three cross-sectional surveys conducted in 2011 (baseline), 2015 (first follow-up) and 2019 (second follow-up). Indicators of health outcomes and determinants of health were assessed in children and women, and at the household-level, in both project-impacted and non-impacted comparison communities.

## 2. Materials and Methods

### 2.1. Ethical Considerations

Study protocols and survey tools were approved by the ethics review committee of the Tropical Disease Research Centre in Ndola, Zambia (TRC/ERC/04/07/2011, TRC/C4/07/2015 and TRC/C4/01/2019). Communities were sensitised in advance to the objectives and activities of the surveys. A written informed consent (i.e., signature, or fingerprint for illiterate individuals) was obtained from participating women aged 15–49 years for themselves and their children <5 years. In primary schools, parents were asked to provide prior informed consent to participate in the survey, with head teachers supporting the individual consent process as a proxy for participating school attendees aged 9–14 years.

Results of biomedical tests were obtained and communicated to the participants immediately or as soon as the microscopic analyses were completed. In line with national guidelines, children with *Plasmodium falciparum* infections were treated with an artemisinin-based combination therapy. Children and women with mild to moderate anaemia (i.e., haemoglobin (Hb) 7–11 g/dL) were given iron supplements and advised to follow up at a public health facility. Children and women with severe anaemia (Hb <7 g/dL) were referred to a public health facility for further care. Women infected with *Treponema pallidum* were given a stat dose of 2 g azithromycin, plus treatment for their sexual partner(s). All participating school attendees received treatment with albendazole (400 mg) and those with confirmed *Schistosoma* infection received praziquantel (40 mg/kg) following guidelines of the World Health Organization (WHO) [11]. All treatment was provided free of charge to the study participants.

### 2.2. Study Area

The project is located in a previously forested area in the Kalumbila district, North-Western province, bordering the Democratic Republic of Congo in the north (Figure 1). The native host population were predominantly subsistence farmers of low socio-economic status [6]. The project covers an area of 950 km^2^ and infrastructural changes since its development included building of the mining infrastructure (e.g., open pit mine, processing plants), roads, an airstrip, two large dams and a game conservation area. Several communities were resettled due to the project development and new settlements were established [7,8]. Considerable labour-seeking in-migration of people resulted in urbanisation of several communities.

### 2.3. Study Design and Sampling

The three cross-sectional surveys were conducted in 2011 (June/July), 2015 (July) and 2019 (June/July). While the 2011 survey is considered the baseline before project development, the subsequent surveys are considered as follow-ups to monitor and compare changes with the pre-project situation. The surveyed communities, selected through semi-purposive sampling, included nine impacted communities considered directly affected by the project (e.g., by resettlement, project induced in-migration and labour source) or who benefit from the project-supported health interventions, and four comparison communities (Figure 1) [7,8,12]. Comparison communities were defined as neither directly impacted by the project nor having received any project-initiated health intervention [7,8]. Within the communities, a quota sampling of between 25 and 35 randomly selected households was performed. In order to increase representativeness in larger communities (i.e., Chisasa, Musele and Kanzala), the sample quota was doubled. The presence of at least one woman and one child <5 years were the inclusion criteria. For recruitment of school attendees, all primary schools serving the 13 selected communities were included. Children were selected randomly among stratified groups to achieve an even spread between age-groups (9–10 years, 11–12 years and 13–14 years) and across gender.

### 2.4. Data Collection

Data collection was conducted using three survey modules [12]: (i) a questionnaire survey for women of reproductive age (15–49 years); (ii) an assessment of biomedical indicators in children aged <5 years and women aged 15–49 years; and (iii) an assessment of intestinal parasites and schistosomiasis in school attendees aged 9–14 years. The questionnaire investigated household characteristics, demographic and socioeconomic characteristics, as well as knowledge, attitudes, behaviours and practices related to health. Questionnaire data were collected using Open Data Kit (ODK) on tablet devices. Devices were password protected and data were stored on a server at the Swiss Tropical and Public Health Institute (Swiss TPH; Basel, Switzerland) and encrypted with a secure sockets layer. After the questionnaires were administered, the height and weight of children <5 years and women were measured following WHO anthropometric guidelines [13]. Infection with *P. falciparum* in children 6–59 months was assessed using a rapid diagnostic test (RDT; SD Bioline Malaria Ag Pf Rapid Test, Standard Diagnostics, Macmed Healthcare Ltd.; Nairobi, Kenya). The HemoCue^®^ test (HemoCue AB; Ängelholm, Sweden) was used to assess Hb levels in women and children (6–59 months). Syphilis testing in women was done using the Alere Determine^TM^ Syphilis TP antibody test (Abbott; Abbott Park, Chicago, IL, USA).

In each of the 13 selected schools, stool and urine samples were collected from 30 or 60 children (depending on community sample size) between 10 a.m. and 2 p.m. when maximum egg excretion has been shown to occur, thereby increasing the likelihood of detecting *Schistosoma* infections [14]. The urine samples were inspected for visible haematuria and examined on the same day by the 10 mL sedimentation method for presence of *Schistosoma haematobium* eggs. The stool samples were examined for helminths, including *S*. *mansoni*, *Ascaris lumbricoides*, *Trichuris trichiura* and hookworm spp., within 24 h of collection by the Kato Katz method using a 41.7 mg template (Vestergaard Asia, PVT Ltd.; Delhi, India) [15].

### 2.5. Data Analysis

Descriptive analyses were performed to compare frequencies, means and 95% confidence intervals (CI) across survey years and project impact (impacted vs. comparison communities). Interactions between years (2011 vs. 2019) and the project’s impact were reported using odds ratios (ORs) with corresponding 95% CI and *p*-values.

We used an inventory of household assets and amenities, similar to the one developed by the Demographic and Health Survey (DHS), to measure levels of household wealth [16]. Our measure used a reduced set of the most important inventory items: number of household members per sleeping room; source of drinking water; type of floor, roof and wall material; type of cooking fuel; and ownership of key household assets (e.g., radio, television, bicycle, phone and bank account). Principal component analysis (PCA) was used to create a single dimension asset score. The PCA was conducted together with the raw 2007 and 2013–14 DHS data to ensure that our data were nationally comparable. Household wealth quintiles were then computed based on the first principal component of the PCA [17].

Two logistic regression models were built in order to (i) estimate the changes of *P. falciparum* infection in children 6–59 months in impacted vs. comparison communities over time; and (ii) assess the risk of acquiring malaria (outcome) when living in an impacted community as compared with comparison communities (primary predictor) after adjusting for other explanatory variables. The model was adjusted for access to health care (i.e., presence of a health facility in the community), household years of residency in the area (≤10 years vs. >10 years), household resettlement for the project (yes vs. no), structure of housing (rudimentary vs. solid), household wealth index, indoor residual spraying (IRS) in the 12 months preceding the survey (yes vs. no), household having received malaria community outreach active case finding (“malaria seek and treat”) in the 12 months preceding the survey (yes vs. no), paid employment of at least one household member (yes vs. no), mother’s level of education, mother’s consistent knowledge on malaria (yes vs. no) [18] and children’s use of bednets in the night preceding the survey (yes vs. no). Odds ratios (ORs) with 95% CI and *p*-values were reported for both models.

In order to determine the nutritional status of children <5 years of age, moderate levels (i.e., levels between −3 and −2 z-scores below the median standards) of wasting (i.e., low weight-for-age or acute malnutrition), stunting (i.e., low height-for-age or chronic malnutrition) and undernutrition (i.e., low weight-for-height) were calculated according to WHO child growth standards [13].

Statistical analyses were performed using Stata version 15.0 (Stata Corporation; College Station, USA) and R version 3.4.3 (The R Foundation; Vienna, Austria).

## 3. Results

### 3.1. Study Population

Sample sizes, stratified by the cross-sectional surveys, community and impact status, are summarised in Table 1.

### 3.2. Vector Control and Malaria Indicators

Bednet use among children aged 6–59 months increased from 28.0% (95% CI 23.9–32.4%) in 2011 to 52.5% (95% CI 49.1–56.0%) in 2019, with similar trends in impacted and comparison communities (Figure 2 and Appendix A). Data on IRS were not collected in 2011, as it was known that no such intervention was being conducted at that time. Whilst in 2015, a significant gap existed between the impacted (69.8%; 95% CI 64.7–74.5%) and comparison communities (28.9%; 95% CI 21.1–37.6%), the IRS rate in the comparison communities improved significantly in 2019 (88.6%; 95% CI 82.6–93.1%), with the coverage in impacted communities being similarly high at 81.8% (95% CI 77.3–85.7%).

The overall prevalence of *P. falciparum* infection in children aged 6–59 months increased from 19.0% in 2011 (95% CI 15.5–22.8%) to 38.2% in 2015 (95% CI 34.7–41.7%) and 45.6% in 2019 (95% CI 42.1–49.0%). The results in Table 2 show that (i) the odds of being infected with *P. falciparum* in 2019 were 2.65 (95% CI 1.33–5.27) times higher than in 2011; (ii) the impacted communities were overall significantly less affected by *P*. *falciparum* (OR = 0.41; 95% CI 0.21–0.82); and (iii) the odds of infection with *P. falciparum* in an impacted community vs. comparison community did not change significantly over time (OR = 1.18; 95% CI 0.55–2.50).

Results from the logistic regression model show that living in a community impacted by the project significantly lowered the risk of acquiring *P. falciparum* infection (OR = 0.68, 95% CI 0.49–0.94) after adjusting for other factors that potentially influence the probability of infection (Table 3). A low socio-economic status of the household and lower education level of the mother were found as risk factors for a child to become infected with malaria.

### 3.3. Nutritional Indicators and Anaemia in Children Under 5 Years of Age

The prevalence of wasting in both the impacted and comparison communities was generally low (<2.5%) and did not change significantly over time (Figure 3 and Appendix A). While almost half of the children <5 years were stunted in 2011 (48.7%, 95% CI 44.1–53.2%), stunting improved over time (2015: 39.9%, 95% CI 36.7–43.1%; 2019: 30.0%, 95% CI 27.1–33.2%). The proportion of children underweight declined slightly from 12.5% (95% CI 9.7–15.8%) in 2011 to 9.8% (95% CI 8.0–12.0%) in 2019.

Overall, between the baseline and 2019 studies, the prevalence of anaemia in children aged 6–59 months has remained at a concerning level (46.8%; 95% CI 43.3–50.2%). Across all survey years, rates were non-significantly lower in impacted communities compared with non-impacted communities.

### 3.4. Schistosomiasis and Soil-Transmitted Helminth Infections among Children Aged 9–14 Years

With the exception of *S*. *mansoni*, the burden of all parasites declined in the impacted communities since the 2011 survey (Figure 4 and Appendix A). *S*. *mansoni* was 1.1% (95% CI 0.2–3.1%) in 2011 compared with 7.6% (95% CI 5.0–11.0%) in 2019, showing a marked increase. In the comparison communities, the parasite burden reduced between 2015 and 2019 for all investigated parasites, except *S*. *haematobium*, where prevalence has remained unchanged.

In 2019, the prevalence of helminth infections (except for *S*. *mansoni*) was lower in impacted communities compared with comparison communities. The prevalence of both *T. trichiura* and *A. lumbricoides* was lower in 2019 (4.8% and 0.0%, respectively) than in 2011 (7.4% and 0.6%, respectively). The odds of being infected with hookworm in an impacted vs. a comparison community over the study period were significantly lower (OR = 0.27; 95% CI 0.11–0.64).

Taken together, the proportion of uninfected children was markedly higher in 2019 (63.1%; 95% CI 58.6–67.5%) compared with the previous survey periods (27.5%; 95% CI 22.6–32.8% in 2011 and 29.9%; 95% CI 25.7–34.4% in 2015). Additionally, in 2019, the percentage of uninfected children was significantly higher in the impacted (70.3%; 95% CI 65.1–75.2%) than in the comparison communities (47.3%; 95% CI 39.1–55.6%).

### 3.5. Maternal Health Indicators

The percentage of mothers delivering the last-born child in a health facility has increased significantly from 63.9% (95% CI 58.4–69.2%) in 2011 to 95.3% (95% CI 93.3–96.9%) in 2019 (Figure 5 and Appendix A). Prevalence of underweight (i.e., a body mass index < 18.5) in women did not change significantly between 2015 (9.9%; 95% CI 7.6–12.6%) and 2019 (10.5%; 95% CI 8.1–13.2%), with a small increase recorded in impacted communities and a small decrease in comparison communities.

Prevalence of anaemia in women in 2019 (17.6%; 95% CI 14.7–20.9%) was similar to baseline levels measured in 2011 (17.9%; 95% CI 13.8–22.5%). The prevalence of syphilis was slightly higher in 2019 compared with 2015 (4.8%; 95% CI 3.3–6.9% vs. 4.2%; 95% CI 2.7–6.3%). No significant differences were observed between impacted and comparison communities in terms of anaemia prevalence and syphilis among women of reproductive age.

## 4. Discussion

Data obtained from three cross-sectional surveys conducted at 4-year intervals (2011, 2015 and 2019) support the monitoring of health determinants and outcomes in communities potentially impacted by a large copper mining project. Improvements (e.g., use of vector control measures, stunting in children <5 years and hookworm prevalence in children 9–14 years), unchanged prevalence (e.g., anaemia in children and women and syphilis prevalence in women) and deteriorations (e.g., *P. falciparum* prevalence in children 6–59 months) of health indicators were observed. In addition, we found evidence that living in a village impacted by the project provided a protective effect against some diseases (e.g., *P. falciparum* infection in children aged 6–59 months and hookworm infection in children aged 9–14 years). The findings are contextualised and discussed in the following sections.

### 4.1. Vector Control and Malaria Indicators

Despite certain limitations for comparability (e.g., sample size, local heterogeneities and seasonal variations), the increasing *P. falciparum* malaria prevalence found in the study area coincided with observations at the provincial level. Indeed, the prevalence measured during the Malaria Indicator Surveys (MIS) in the North-Western province was 17.3% (2010), 32.5% (2012), 40.6% (2015) and 36.3% (2018) as compared with overall 19.0% (2011), 38.2% (2015) and 45.6% (2019) in the study area [7,19,20,21,22].

In our study population, apart from the apparent significant increase in malaria point prevalence in both the impacted and comparison communities, living in a community considered impacted by the project was seen as a protective factor against a child’s risk for contracting malaria, whilst the low socio-economic status of the household and lower educational level of the mother were noted as risk factors. Persistent malaria will continue to result in school absenteeism and lower productivity and thus further advance the very same risk factors for malaria infection. These results confirm the notion of the interplay between poverty, economic development and malaria [23,24]. They also highlight the special attention required to assist the most vulnerable population groups that, in the study area, include the poorest households and the native host population that has not been directly impacted by the mining project [7].

Looking beyond the study area, the *P. falciparum* infection rate has significantly increased despite the tremendous scaling up of control strategies. Initial successes in malaria control, especially in the northern parts of Zambia, seem to have waned in recent years [25,26]. Potential reasons for this and previous resurgences include the emergence and spread of resistance to insecticides and anti-malarial drugs, natural variations in mosquito populations and attenuations or interruptions of control activities, often linked to reduced funds [27]. Given the consistently increasing trend, both at the local and provincial level, a thorough investigation of the current control efforts (including strategies, efficacy of interventions and surveillance) as well as entomological studies (e.g., abundance and composition of mosquito populations and insecticide sensitivity) are indicated.

### 4.2. Nutritional Indicators and Anaemia in Children Under 5 Years

Findings from the surveys indicate that acute malnutrition (i.e., wasting) was not a pressing public health problem with overall rates below 2% across the various periods. Wasting rates in the communities around the mining project were consistently lower than what was found in North-Western province in the repeated DHS (8.2% in 2013–14 and 2.4% in 2018) [18,28]. Regarding chronic malnutrition (i.e., stunting), findings were encouraging over the surveyed years, with significant decreases achieved since the 2011 baseline (−18.7% overall). Similarly, the prevalence of stunting in the North-Western province decreased steadily as revealed in the repeated DHS (i.e., 43.6% in 2007, 36.9% in 2013–14 and 31.9% in 2018). Hence, successes in the fight against chronic malnutrition have been achieved at the provincial level, beyond the project’s area of influence. Nonetheless, nearly a third of the children below 5 years of age are still below standards for a normally nourished population [13]. Sustained efforts are required to support continued improvement in child nutrition [29].

The high prevalence of anaemia in children 6–59 months (i.e., 46.8% in 2019) remains a considerable public health concern. Hb levels were also measured in several DHS and MIS, with anaemia prevalence found in the study area in 2015 and 2019 still substantially lower than what was found at the provincial level (70.9% in the 2018 MIS and 61.6% in the 2018 DHS) [19,28]. In the North-Western province, the predominant reasons for the high anaemia prevalence are (i) low consumption of iron due to limited variety in their diet; (ii) high rates of infectious disease, including malaria and diarrhoea; and (iii) limited sanitation services leading to high rates of bacterial and parasitic infections, especially hookworm [30]. Hence, the altered epidemiology of infectious diseases, access to healthcare and diets that potentially changed due to the project development, might have influenced rates of anaemia in the study area [31].

### 4.3. Schistosomiasis and Soil-Transmitted Helminth Infections in Children Aged 9–14 Years

The overall burden of intestinal parasites and *S. haematobium* infection in school attendees has decreased significantly given that the proportion of children with no parasites increased from 27.5% (2011) to 63.1% (2019). According to information from the Kalumbila health district, regular de-worming programmes in schools are implemented across the entire district [32]. The consequence of these interventions was noticeable in 2019 given the overall improvements and especially the large decrease in hookworm infections and the absence of *A*. *lumbricoides* infections. Interestingly, in 2019, significantly more children were uninfected in the impacted than in the comparison communities (70.3% vs. 47.3%). In the impacted communities, since 2015, schoolchildren were regularly dewormed with albendazole through FQML’s mother and child health, supported by water, hygiene and sanitation initiatives. These private sector support initiatives, including deworming, supporting access to safe sanitation (32.9% in impacted vs. 5.0% in comparison communities; unpublished data from the questionnaire survey) and potential changes due to economic development, such as wearing shoes, might be responsible for this difference [33,34].

Persistent infections, however, indicate that the environment (i.e., water and soil) is still contaminated. For both schistosomiasis and soil-transmitted helminthiases, a holistic control approach combining regular treatment, the use of safe sanitation and behavioural changes (e.g., no open defecation and urination) is indicated in this setting to gain and sustain control [32]. It is to be expected that the definitive interruption of transmission will only occur with significant improvements in socio-economic circumstances, since these are ultimately poverty-driven diseases.

### 4.4. Maternal Health Indicators

The rate of underweight in women in our surveys (9.9% in 2015 and 10.5% in 2019) is comparable with observations elsewhere in the North-Western province (9.2% in 626 individuals), as observed during the 2013–14 DHS [18]. Anaemia levels were, however, consistently lower in the study area (17.9%, 25.0% and 17.6% in 2011, 2015 and 2019, respectively) compared with levels reported in other areas of the province (32.2% during the 2018 DHS) [28]. The differences could be explained by various factors given the diverse aetiology of anaemia. The findings of the survey indicate that almost every one in five women remain anaemic in the study area and reasons may relate to persistent micro-nutrient deficiencies, inherited disorders and sustained parasitic infections (e.g., hookworm and malaria) [31]. The reported increase in malaria prevalence in children in the current survey may also play a role in maternal anaemia. There is an opportunity to support micro-nutrient and iron supplementation in women in the study area to reduce the potential impacts of anaemia on maternal and child health [35].

The slight, non-significant increase in the overall syphilis prevalence in the impacted communities between 2015 and 2019 is an important finding, considering that this is a setting regarded as high risk for the transmission of sexually transmitted infections (STIs) [36]. This unchanged trend probably reflects the extensive STI and HIV interventions that FQML support in the study area (identified as a priority in the HIA) [36]. It is evident that concerted efforts are needed to turn the slight increase into a declining trend. For instance, syphilis testing capacities in local health facilities could be improved if RDTs were more readily available (only 6 out of the 11 health facilities in the surveyed communities had RDTs).

### 4.5. HIA to Monitor Community Health Impacts of Mining Activities

HIA should, as per definition, guide the establishment of a framework for monitoring and evaluation of health impacts [10]. A community health management plan, including intervention and monitoring recommendations, was one of the concrete outcomes of the initial assessment in 2010 that has followed HIA best practice principles and industry good practice guidelines [9,10]. To complement the community health management plan, environmental surveillance of air, soil and water in the project area is done by the project on a continuous basis to detect eventual health hazards in a timely manner. Even though direct causality cannot be inferred with the here-presented cross-sectional modular survey data on standardised health indicators, the applied methodology has several meaningful benefits for the current setting. First, it allows for the assessment of identified health impacts and supports the reassessment of those impacts. Secondly, it allows for the identification of trends, disease hotspots and vulnerable populations [37]. Thirdly, it is useful for adjustment and tailoring of health interventions in specific communities. Fourthly, it allows for comparison of findings in project-impacted communities with comparison communities surveyed in parallel as well as with the wider province.

Most large-scale natural resource extraction projects in sub-Saharan Africa are implemented in rural settings where health data are notoriously scarce, and the implementation of HIA is usually not required or regulated by the host country [38,39]. For these and any other development projects, HIA can play a pivotal role in protecting, promoting and monitoring the health of populations that are affected by such projects implemented outside the health sector [40]. In order to eventually assess an HIA’s promise to minimise potential negative effects, enhance potential benefits and work towards sustainable development, the HIA best practice and guiding principles, especially those relating to monitoring and transparency, need to be applied by an increasing number of project proponents [39].

### 4.6. Limitations

Our study has several limitations. First, the semi-purposive, non-random community sampling limits the generalisation of results to non-surveyed communities. Secondly, for several communities and a few indicators (e.g., syphilis, IRS), no data were available for the baseline survey in 2011. Thirdly, this study was not a longitudinal study measuring the same indicators in the same participants. It may be that the representativeness of the survey varied over the three survey rounds. However, that risk is minimized by following the same random sampling methods over the three surveys and maintaining representative coverage of households within each village. Fourthly, the prevalence of *P. falciparum* has the potential to be slightly overestimated as previously treated children may test positive, up to several weeks later, for the parasite antigen detected by the RDT [41,42]. Fifthly, similarly for syphilis, the prevalence might be slightly overestimated as the RDT detects both active and past infection [43]. Sixthly, the prevalence of *S. mansoni*, *S. haematobium* and soil-transmitted helminth infections are likely underestimated as only single stool and urine samples were examined microscopically [44].

## 5. Conclusions

Changes in health indicators observed in the study area coincided largely with observations from the North-Western province of Zambia. However, communities impacted by the mining project frequently showed better health outcomes than comparison communities. In the current setting, both the interplay between health and economic growth and the health interventions promoted by the project’s HIA might be reasons for the observed better health status in the impacted communities, compared with comparison communities [45]. Caution is indicated to avoid promotion of health inequalities within and beyond the project area, especially for native host communities that might not have sufficient capacity to adapt to the rapidly transforming setting.

## Figures and Tables

**Figure 1 ijerph-17-03633-f001:**
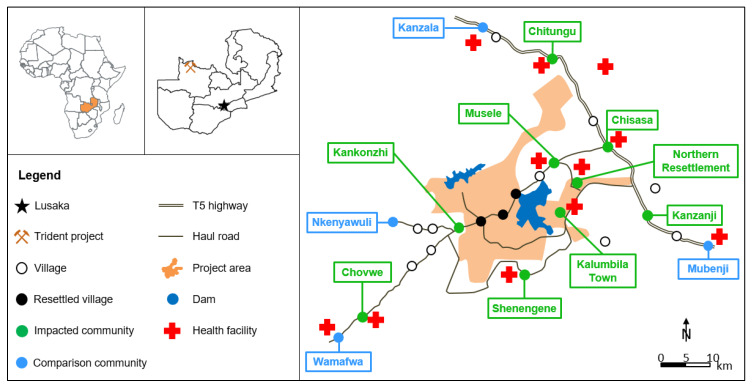
Map of the study area, Kalumbila district, Zambia.

**Figure 2 ijerph-17-03633-f002:**
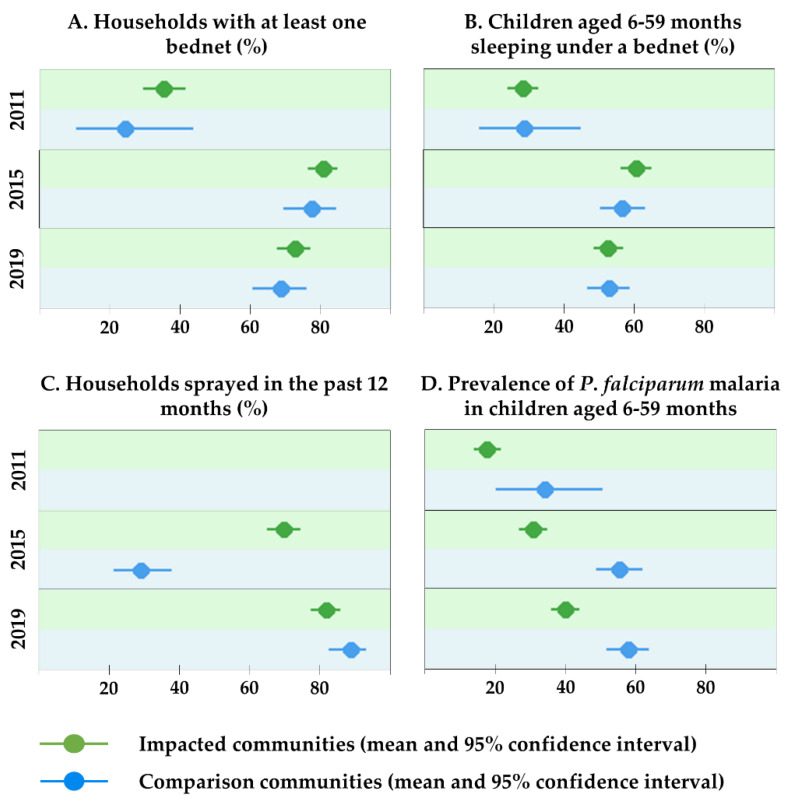
Vector control and malaria-related indicators, Trident project, Zambia (2011, 2015, 2019).

**Figure 3 ijerph-17-03633-f003:**
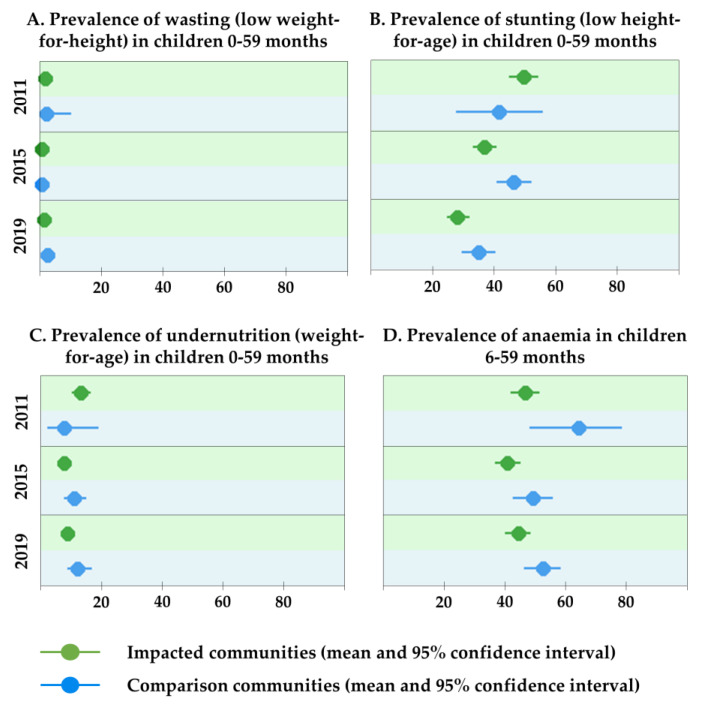
Nutritional indicators and anaemia, Trident project, Zambia (2011, 2015 and 2019).

**Figure 4 ijerph-17-03633-f004:**
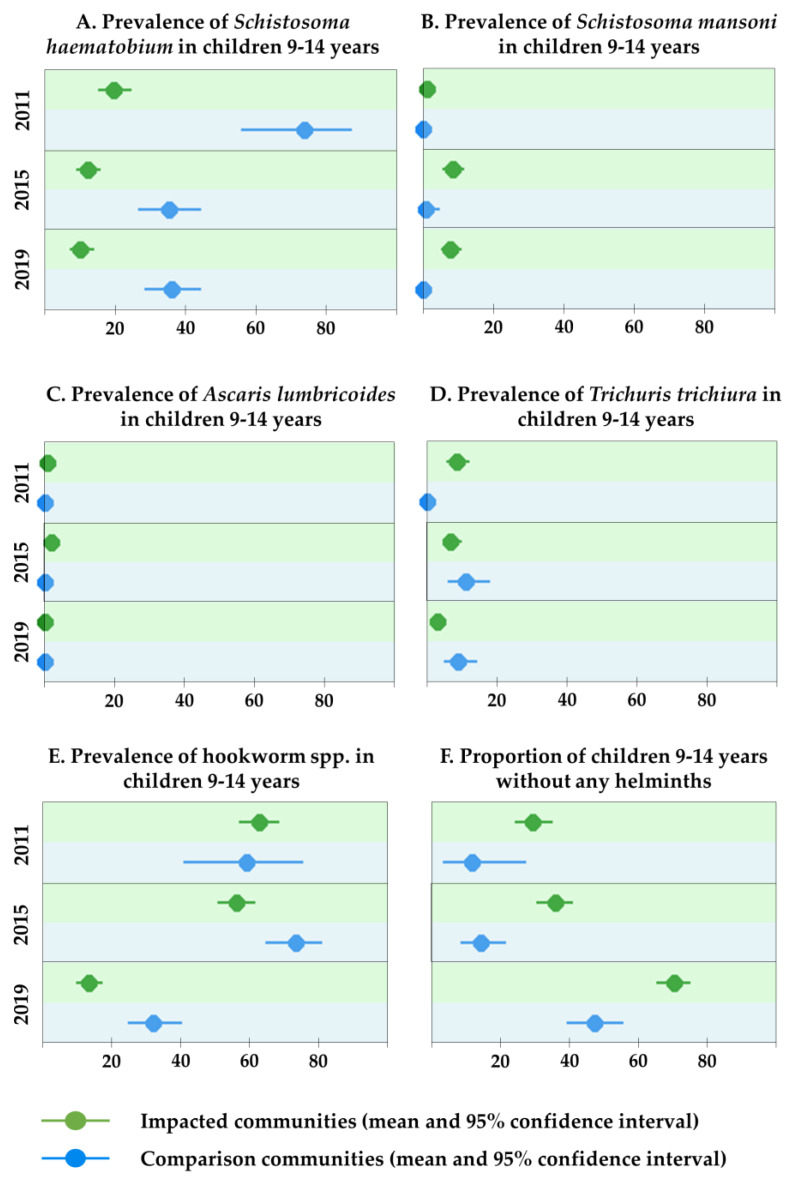
Schistosomiasis and soil-transmitted helminth indicators, Trident project, Zambia (2011, 2015 and 2019).

**Figure 5 ijerph-17-03633-f005:**
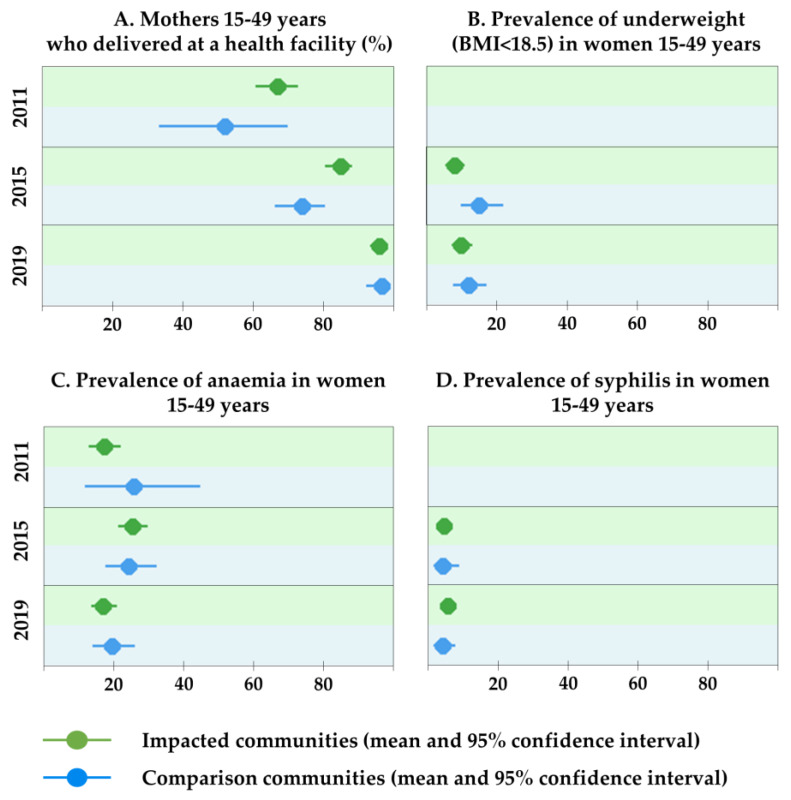
Women’s health indicators, Trident project, Zambia (2011, 2015 and 2019).

**Table 1 ijerph-17-03633-t001:** Study population, Trident project, Zambia (2011, 2015 and 2019).

Community	Number of Households			Children Aged <5 Years			Children Aged 9–14 Years			Females Aged 15–49 Years		
	2011	2015	2019	2011	2015	2019	2011	2015	2019	2011	2015	2019
Impacted communities												
Kalumbila Town ^1^	n/s	29	31	n/s	43	39	n/s	30	30	n/s	31	32
Wanyinwa ^2^/Northern Resettlement ^1^	35	34	30	64	70	53	35	30	30	42	44	36
Shenengene ^1^	n/s	32	31	n/s	52	58	n/s	30	30	n/s	35	34
Musele	30	67	65	48	134	128	40	59	60	30	95	86
Chisasa	62	65	66	97	112	111	44	60	60	72	77	77
Kankonzhi	39	30	32	73	58	56	35	30	30	48	39	39
Chovwe	63	32	32	95	54	51	66	30	30	60	38	37
Kanzanji ^3^	n/s	32	32	n/s	59	59	n/s	29	30	n/s	38	37
Chitungu	30	33	32	59	51	56	59	30	30	30	38	38
**Total**	**259**	**354**	**351**	**436**	**633**	**611**	**279**	**328**	**330**	**282**	**435**	**416**
Comparison communities												
Nkenyawuli	29	32	31	51	67	62	34	30	30	31	32	39
Wamafwa ^3^	n/s	33	32	n/s	73	55	n/s	30	30	n/s	38	35
Kanzala ^3^	n/s	30	63	n/s	56	119	n/s	30	60	n/s	35	70
Mubenji ^3^	n/s	33	32	n/s	61	59	n/s	30	30	n/s	43	41
**Total**	**29**	**128**	**158**	**51**	**257**	**295**	**34**	**120**	**150**	**31**	**148**	**185**

^1^ Newly developed community after 2011 [7]. ^2^ 97% of residents in Northern Resettlement originated from Wanyinwa (2015 data) [7]. ^3^ Surveyed community added in 2015 to increase sample size [7]. n/s, not sampled.

**Table 2 ijerph-17-03633-t002:** Logistic regression model comparing the prevalence of *P*. *falciparum* in children aged 6–59 months before (2011) and 8 years after implementation of the project (2019) in impacted and non-impacted comparison communities, Trident project, Zambia.

	OR (95% CI)	*p*-Value
After project implementation (2019) vs. before (2011)	2.65 (1.33–5.27)	0.006
Impacted vs. comparison communities	0.41 (0.21–0.82)	0.011
Interaction between year and impact status	1.18 (0.55–2.50)	0.672

**Table 3 ijerph-17-03633-t003:** Logistic regression model comparing *P. falciparum* prevalence in children aged 6–59 months in impacted vs. non-impacted comparison communities with inclusion of secondary predictors, Trident project, Zambia (2011, 2015 and 2019).

	OR (95% CI)	*p*-Value
Community is impacted by the project	0.68 (0.49–0.94)	0.021
Community has local health facility	0.69 (0.45–1.07)	0.097
Household in area for >10 years	1.21 (0.86–1.71)	0.264
Household has been resettled	0.61 (0.34–1.10)	0.101
Household has solid housing structures	0.93 (0.57–1.51)	0.764
Household wealth index		
Lowest	3.99 (1.07–14.94)	0.040
Second	2.05 (0.75–5.58)	0.160
Middle	2.21 (1.00–4.92)	0.051
Forth	1.79 (0.85–3.80)	0.128
Highest	1.00 (reference population)	
Household sprayed with insecticide in the 12 months preceding the survey	1.06 (0.73–1.54)	0.763
Household received “malaria seek and treat” intervention	1.14 (0.82–1.58)	0.432
Household with at least one member employed	0.99 (0.63–1.57)	0.975
Mother’s educational level		
No education or some primary	2.67 (1.31–5.43)	0.007
Primary schooling	1.89 (0.93–3.81)	0.077
Secondary schooling or higher	1.00 (reference population)	
Mother has consistent knowledge on malariatransmission ^1^	0.82 (0.69–1.12)	0.205
Child slept under bednet the night preceding the survey	0.67 (0.57–1.04)	0.083

CI, confidence intervals; n/a, not applicable; OR, odds ratio. ^1^ Knowing that being bitten by mosquitoes is the only true mode of malaria transmission [19].

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
