# Peer review of "Community Health Impacts of the Trident Copper Mine Project in Northwestern Zambia: Results from Repeated Cross-Sectional Surveys"

_ijerph, 2020, doi:10.3390/ijerph17103633_

Round 1

Reviewer 1 Report

  [IJERPH] Manuscript ID: ijerph-797570

Paper titled: Community health impacts of the Trident Copper Mine project in  northwestern Zambia: results from repeated cross-sectional surveys

Dear authors,

I wish to congratulate you for the work you have done, which I found well articulated, sufficiently explained, with a statistical analysis suitable for the intended purposes, a sufficiently reasoned discussion and therefore generally acceptable.

My concern is that the mining activity, also stressed in the title, remains very much in the background because no evaluations are carried out on the variables covered by the HIA. The work is therefore aimed at verifying the improvements, especially in terms of infectious diseases and some health services, in mine areas vs reference areas, without a link with the results of the HIA, which is, however, little explained.

The results are therefore in favor of better prevention in mine areas focused on infectious diseases, evidently as mitigation of the risks attributable to mining activities, recognized for the population but of which no mention is made.

Some minor aspects:

1/16 line 29: WHERE “The prevalence of Plasmodium falciparum infection increase ?

2/16 line 57: The main objective of the HIA were….. I imagine that the first was to estimate the main potential impacts on health

2/16 line 89: it seems that the three surveys were conducted on different people, if yes how possible temporal changes in the compared populations were controlled for ? or in other words, was it possible to verify the representativeness of the samples with respect to the reference populations in the three cross.sectional cuts ?

Error! Reference source not found in several p

Author Response

Comments by Reviewer #1:

Dear authors,

I wish to congratulate you for the work you have done, which I found well articulated, sufficiently explained, with a statistical analysis suitable for the intended purposes, a sufficiently reasoned discussion and therefore generally acceptable.

R1: We thank Reviewer #1 for the overall positive appraisal of our manuscript.

My concern is that the mining activity, also stressed in the title, remains very much in the background because no evaluations are carried out on the variables covered by the HIA. The work is therefore aimed at verifying the improvements, especially in terms of infectious diseases and some health services, in mine areas vs reference areas, without a link with the results of the HIA, which is, however, little explained.

The results are therefore in favor of better prevention in mine areas focused on infectious diseases, evidently as mitigation of the risks attributable to mining activities, recognized for the population but of which no mention is made.

R2: As per definition, HIA has the goal to cover the full range of potential health effects of a given project on an affected population [1, 2]. The indicators (or ‘variables’) assessed (or ‘evaluated’) in the HIA are therefore numerous. However, for the current manuscript, we present a selection of community health indicators, as stated in line 69 of the manuscript. Hence, this manuscript does cover indicators identified as priority health issues by the HIA. A more detailed description on the rational of selection of these specific indicators are given in two preceding manuscripts published on this project [3, 4].

In addition, the focus on infectious diseases is justified by their importance in the current setting. In Zambia, 45% of disability-adjusted life years (DALYs) in 2016 were due to infectious diseases [5]. The importance of infectious diseases was also recognized in the HIA, whereas the true burden of disease in the communities is not captured by the routine health information system. For example, the assessment of the true prevalence of malaria, intestinal parasite infections, nutritional status or syphilis warrant community-based surveys.

Importantly, the cross-sectional surveys did not serve the purpose of ‘verifying the improvements’. The purpose of the surveys was to monitor internationally standardized indicators and the empiric results showed that there were positive trends (‘improvements’) as well as negative trends or no changes at all for specific indicators.

The link of the cross-sectional surveys to the ‘mining activity’ (= the project) was established through our study design, distinguishing impacted and comparison communities. Thus, all results are presented in the light of the presence or absence of the influence of the ‘mining activity’. Hence, we feel that the link to the ‘mining activity’ is prominently featured throughout the manuscript. Of note, the HIA covers health impacts in the affected communities, i.e. ‘outside the fence’, as opposed to the occupational health risk assessment ‘inside the fence’ that would cover the health of mine workers directly involved in the ‘mining activities’.

While the above explanations should help to clarify the points raised by Reviewer #1, we have nevertheless edited and expanded on the introduction section in response to Reviewer #1’s comments. The revised narrative in lines 58-68 reads as follows:

“FQML commissioned a health impact assessment (HIA) in 2010, prior to the development of the mine. The goal was to anticipate and manage potential health impacts on communities in proximity to the project [3, 4, 6]. The HIA applied a determinants of health model – comprising the social, economic and physical environment and individual characteristics and behaviours – to cover the full range of potential direct and indirect health effects of the project on the population [1, 7]. The specific objectives of the HIA were to (i) prevent negative health impacts; (ii) promote health opportunities; (iii) develop a community health management and monitoring plan; and (iv) act on monitoring data to adjust health interventions over time [1]. The monitoring plan included both collection of routine health information system data as well as periodic, cross-sectional survey data for indicators that were considered potential determinants for human health within the HIA.”

Some minor aspects:

1/16 line 29: WHERE “The prevalence of Plasmodium falciparum infection increase?

R3: We thank Reviewer #1 for this important observation. We have reworded to enhance clarity: “The prevalence of Plasmodium falciparum infection increased in both the impacted and comparison communities between 2011 and 2019 but remained consistently lower in the impacted communities.” (see lines 32-34 of the revised manuscript).

2/16 line 57: The main objective of the HIA were….. I imagine that the first was to estimate the main potential impacts on health

R4: Indeed, that was the overarching goal of the HIA as described in the previous sentence: “In order to anticipate and manage potential health impacts on communities living in proximity to the project, FQML commissioned a health impact assessment (HIA) in 2010, prior to the development of the mine [3, 4, 6].” However, to enhance clarity and expand on the rationale based on the comments put forth by Reviewers #1 and #3, we have revised this sentence (see also R2).

2/16 line 89: it seems that the three surveys were conducted on different people, if yes how possible temporal changes in the compared populations were controlled for? Or in other words, was it possible to verify the representativeness of the samples with respect to the reference populations in the three cross-sectional cuts?

R5: This is a good point. Reviewer #1 is correct, as we pursued three cross-sectional surveys rather than a longitudinal study of the same participants, which bears the risk of study population differences. The three cross-sectional surveys provide snapshots of community health at the three specific time points. While there may have been random differences in the representativeness of the study population across the three surveys, these differences should be small, as the recruitment of households was conducted in the same way in each of the surveys and a representative sample size was sampled.

At the same time, the study setting is characterised as a dynamic environment, where demographic changes were expected and indeed took place. The survey tools – specifically the questionnaire survey – had the objective to capture such demographic and socio-economic background changes and these found application in our statistical models (e.g. wealth index, educational attainment and employment).

In conclusion, we agree that this limitation should find consideration in our manuscript and we have added the following points to the limitations section: “Third, this study was not a longitudinal study measuring the same indicators in the same participants. It may be that the representativeness of the survey varied over the three survey rounds. However, that risk is minimized by following the same random sampling methods over the three surveys and maintaining representative coverage of households within each village.” (see lines 395-399 of the revised manuscript).

Error! Reference source not found in several p

R6: Cross-references to Tables and Figures were apparently not correctly updated when the PDF was created. We have corrected the issue and all cross-reference now display correctly.

Reviewer 2 Report

I think it's unclear about how/what the authors tested for consistency of the surveys. For exemple using the Cronbach’s coefficient because it's a statistical measure of internal consistency estimate of reliability of test scores for a given sample.

It would also be very interesting can observer the surveys if it is possible. 

Author Response

Comments by Reviewer #2:

I think it's unclear about how/what the authors tested for consistency of the surveys. For exemple using the Cronbach’s coefficient because it's a statistical measure of internal consistency estimate of reliability of test scores for a given sample.

R7: The quality of the survey conducted is an important point. All three modules (questionnaire, assessment of biomedical indicator and assessment of intestinal parasites and schistosomiasis) used internationally standardized indicators. We used only validated questions from the Demographic and Health Surveys (DHS). Additionally, some items such as household characteristics can be assumed to be highly reliable without formally testing and calculating internal validity (e.g. age and number of children). The assessment of biomedical indicators was done according to international and national guidelines and/or manufacturers protocols. These measurements and tests have all undergone previous validity testing. No further testing of internal validity was performed.

It would also be very interesting can observer the surveys if it is possible. 

R8: We are afraid we do not understand this point offered for consideration by Reviewer #2. Hence, no action was taken.

Reviewer 3 Report

  1. Abstract keywords: Words used in the title cannot be repeated as keywords.
  2. The rationale of the introduction section is unacceptable. For introduction, it is important to review the health impact in the other nations that have similar conditions with Zambia in socioeconomic and natural resources. A summary of pertinent findings and literature citation in each paragraph of introduction should be provided.
  3. There are many “Error! Reference source not found” in the manuscript. What happen? Please check it!
  4. In discussion section, author wrote “the project provided a protective effect against some diseases”. However, no relevant descriptions about trident copper mine project are presented through the manuscript. I would suggest to add contents of the project in the materials and methods section would provide more solid evidence of protective

Author Response

Comments by Reviewer #3:

Abstract keywords: Words used in the title cannot be repeated as keywords.

R9: After consultation with the journal guidelines and the editor, we confirm that keywords can include words used in the title. Hence, we refrained from making any changes to the keywords.

The rationale of the introduction section is unacceptable. For introduction, it is important to review the health impact in the other nations that have similar conditions with Zambia in socioeconomic and natural resources. A summary of pertinent findings and literature citation in each paragraph of introduction should be provided.

R10: We understand the point raised by Reviewer #3; yet, the current manuscript does not cover comparisons to other countries for several reasons. Our position is justified as follows: First, we tried our level best to summarize the literature on this in other countries but such evidence of health effects of large-scale mining activities on population health is mostly anecdotal because (i) primary health data is rarely collected and hence, not documented; and/or (ii) the data are not publicly available. This was also reported in a recent publication by Brisbois et al. (2017) [8]. References reporting on the potential detrimental effects of mining activities on community health are nevertheless given (see lines 49-51 of the revised manuscript). Second, if comparisons to other countries were addressed in the introduction, it should be a running theme throughout the manuscript, specifically in the discussion. Consequently, comparing results found for this large-scale mining setting in Zambia with other settings will be strongly limited by lack of knowledge on the other ‘similar’ settings (e.g. on the health system, prevalent disease patterns, socio-economic and environmental factors, mining regulations and policies with regards to community health, regulations with regards to HIA, etc.). Third, our focus was to put data from mining-impacted communities in relation to available data from nearby, more comparable settings, which in our case were: (i) comparison communities and (ii) the broader North-Western province.

Taken together, we believe that a review of literature of the health impacts is of limited added value and beyond the scope of this paper, also taking in consideration manuscript length.

There are many “Error! Reference source not found” in the manuscript. What happen? Please check it!

R11: The issue has been corrected (see also R6).

In discussion section, author wrote “the project provided a protective effect against some diseases”. However, no relevant descriptions about trident copper mine project are presented through the manuscript. I would suggest to add contents of the project in the materials and methods section would provide more solid evidence of protective

R12: We agree that the manuscript could gain value by providing more information on the project and study area. Hence, we have added a separate sub-chapter entitled ‘2.2. Study area’ to the manuscript (see lines 95-103 of revised manuscript). In addition, the revised narrative in the introduction (see lines 60-63 of revised manuscript) now points out that the “The HIA applied a determinants of health model – comprising the social, economic and physical environment and individual characteristics and behaviours – to cover the full range of potential direct and indirect health effects of the project on the population [1, 7].”, thus implying that effects that arise from the mining activities themselves (direct) as well as secondary, indirect health effects are covered in the HIA. The sentence in question (in full “In addition, we found evidence that living in a village impacted by the project provided a protective effect against some diseases (e.g. P. falciparum infection in children aged 6-59 months, hookworm infection in children aged 9-14 years).”), is backed up by the empiric results and therefore considered appropriate. Importantly, the sentence is implying an association and not a causation as per lines 374-376 in the discussion section.

English language and style:

R13: Based Reviewer #3’s judgement of the English language and style as “Extensive editing of English language and style required”, we had another thorough English proof lecture, sentence-by-sentence, word-by-word. Edits have also been highlighted using a yellow marker.
